# Water Shortage Affects Vegetative and Reproductive Stages of Common Bean (*Phaseolus vulgaris*) Chilean Landraces, Differentially Impacting Grain Yield Components

**DOI:** 10.3390/plants11060749

**Published:** 2022-03-11

**Authors:** Gerardo Tapia, José Méndez, Luis Inostroza, Camila Lozano

**Affiliations:** Unidad de Recursos Genéticos Vegetales, Instituto de Investigaciones Agropecuarias, INIA-Quilamapu, Chillán 3800062, Chile; jose.mendez@inia.cl (J.M.); linostroza@inia.cl (L.I.); camila.lozanov@outlook.com (C.L.)

**Keywords:** drought stress, Mediterranean environment, common bean, landraces, yield traits, water deficit

## Abstract

Water availability for agricultural use is currently a global problem that worsens with climate change in several regions of the world. Among grain legumes, common bean (*Phaseolus vulgaris*) is the most cultivated in the worldwide. The Chilean germplasm of common bean is characterized by tolerance to water stress. Here, we analyzed a selection of nine ancient Chilean landraces in regard to their drought tolerance, simulating optimal (OW) and restricted watering (RW) in a Mediterranean environment. Phenological, growth, and yield traits were recorded, and correlation analysis was performed. Accordingly, leaf temperature and osmotic potential were higher under RW, while the leaf chlorophyll content decreased in all landraces. Physiological maturity days and seed-filling days were lower in RW than in OW. This similarly occurred with the grain yield. The % yield reduction was negatively correlated with the % pod reduction and the relative rate of leaf expansion (RLAE) reduction. However, the 100-seed weight value was not significantly modified by water treatment (*p* > 0.05). For instance, landraces that preferred to fill the grain with a lower rate of leaf expansion showed a lower loss in grain yield under drought conditions. These results suggest that the resource partitioning between growing leaves, flowers, and developing pods in Chilean landraces is variable, affecting the common bean drought tolerance.

## 1. Introduction

Climatic change is already affecting food security in drylands, particularly those in Africa, Asia, and South America. It occurs through increasing temperatures, changing precipitation patterns and increasing frequency in extreme events (IPCC reports). Nearly 21–37% of greenhouse gas emissions come from agriculture and land use, reaching a production of 142 ± 42 TgCH_4_ yr^−1^, 8 TgN_2_O yr^−1^, and 4.9 GtCO_2_ yr^−1^ between 2007 and 2016. Here, the development of sustainable agricultural production and consumption is crucial. Legumes are important for delivering sustainable agricultural production and play central roles: at the food-system level, as a source of protein; at the production system level by their ability to fix atmospheric nitrogen; and at the cropping system level by rotation practices that break pest and disease cycles [1].

Common bean (*Phaseolus vulgaris* L., Fabaceae) is one of the most important grain legumes worldwide [2,3,4,5]. It is a valuable source of dietary protein, fiber, minerals, vitamins, and antioxidants, among other bioactive compounds [3,6]. Common bean is cropped throughout temperate regions where the growing season has a frost-free range of 60 to 120 days, as well as in tropical highlands with growing temperatures <30 °C [3]. Close to 18.9 million tons of common bean are produced worldwide, and the Americas account for 46% of global production.

Mediterranean-like climates are located on all continents, excluding Asia. Mediterranean environments are characterized by hot and dry summers, and wet and mild winters [5]. Mediterranean environments are one of the environments most threatened by climate change [7]. Currently, climate change effects include an uninterrupted sequence of dry years during the last decade with mean rainfall deficits of 20–40% [8]. Nearly 70% of common bean-planted surfaces worldwide are currently threatened by drought, severely affecting grain yield [9,10]. Therefore, common bean drought tolerance genetic improvement has become one of the major challenges for plant-breeding programs worldwide [11].

Historical domestication of local common bean germplasm has driven their adaptation to Mediterranean environments [12]. According to some historical chronicles, selection and exchange of ancestral common bean germplasms were performed by prehispanic cultures [13,14]. In this sense, adapted wild species or crop landraces offer a great source of genetic diversity for developing modern cultivars with improved grain yield under drought conditions [15,16].

In Chile, common bean cultivation is associated with small family farmers who have dedicated small production areas. During the last 30 years, Chilean common bean production has declined because of the increase in cost production, import increase, and displacement by other crops and fruit trees. Additionally, no increase in yield has been obtained due to low plant breeding and technological levels [17].

The common bean germplasm collection preserved in the Chilean seed banks has been collected in a broad geographic distribution, from an arid environment in the north (latitude 18° S) to a cold and humid environment in the south (latitude 42° S) [18]. Morphological and genetic characterization have revealed that they mostly belong to the Chilean race, whose variability pattern has suggested the presence of genetic introgression, showing differences from Andean heritage [19,20]. Thus, Chilean landraces are composed of members of Andean and Mesoamerican pools [21]. Chilean landraces offer an unexplored source of genetic diversity for improving drought tolerance; however, no work has attempted to evaluate them.

Common bean drought tolerance has been broadly studied from physiological and agronomical points of view [22,23,24]. However, most of these studies have been conducted in tropical environments and used Mesoamerican races as the primary source of genes for developing drought-tolerant cultivars [25,26]. Deep rooting offers access to more moisture and greater water status, but it does not result in greater yield [27]. On the other hand, photosynthate partitioning to grain plays a major role in drought tolerance. Partitioning is quantified as the pod partitioning index and pod harvest index. Both parameters consistently present significant correlations with yield under water stress [27]. Recently, it was reported that sink strength plays a key role in the drought tolerance of common bean [28,29]. Sink strength is determined by the ability of a tissue to acquire resources, which relies on resource uptake and metabolism in that tissue. It has been observed that plants that are able to fill seeds under drought maintain the growth of sinks, which correlates strongly with grain yield [28,29].

In the Mediterranean environment, the common bean crop could suffer water stress throughout the entire growing cycle [17]. However, the reproductive stages have been identified as the most sensitive to water stress. In this period, water stress is also accompanied by high temperature (>30 °C), which has detrimental effects on flowering and pod filling [30,31,32]. The aim of this research was to identify the differential traits that rule tolerance to drought in Chilean germplasm and could be associated with the domestication process of selection. Here, we analyzed the phenotypic behavior of nine representative Chilean common bean landraces to determine the genetic variability in drought tolerance and some relevant phenotypic traits involved in the adaptative ability to abiotic stresses. The results account for phenotypic variability in phenology and grain yield traits that are distant from other germplasm sources. This allows us to propose a guideline to improve tolerance to water stress and adaptation in the Mediterranean environment.

## 2. Results

### 2.1. Water Stress and Temperature Effect

This study aimed to determine the behavior and relationship found in common bean landraces under conditions commonly found in Mediterranean field environments, where drought and high-temperature events can occur during crucial moments of plant development that impact crop yield. The contrasting water stress treatment starting at 54 DAS coincided with the flowering state (Figure 1). Water stress was maintained during the entire reproductive period in the RW treatment at a maximum (Figure 1).

Environmental temperature and humidity were recorded in the assay for both treatments. During a period between 54 and 87 DAS, the leaf temperature was measured. Between 60 and 74 DAS, the leaf temperature in the RW treatment was higher than that in the OW treatment (Figure 1b). The change between OW and RW reached an amplitude of 6–7 °C during this period, with a mean of 32–33 °C in the RW treatment. This coincided with the higher difference between soil water availability during this period (Figure 1a). Moreover, this is coincident with a significant increase in the absolute value of Ψπ in RW respect to OW (Figure 1c). During flowering and pod formation, the highest temperature events were recorded, including three times at 40 °C (Appendix A). Previously, at 54 and 87 DAS, the soil water was similar between OW and RW, which coincided with similarities in leaf temperature (Figure 1).

### 2.2. Phenological Traits

With the aims of simulating natural conditions in the field, the plants were subjected to water stress treatment near the start of flowering. The flowering time was significant different among the genotypes (*p* < 0.05); however, no significant differences were found between the two water conditions (*p* > 0.05). Here, the most precocious landraces were Enriqueta, Coyunda, Pinto, Bayote, Tortola, and Manteca, which showed a flowering time of nearly 55 days on average. In contrast, Blanco, Negro argel, and Coscorron were the most significantly delayed flowering landraces, with over 60 to 79 DAS (Figure 2a). Some similarities were maintained during pod development. Here, Coscorron was the most delayed, while Pinto, Coyunda, Bayote, Enriqueta, and Tortola were the earliest. Although Manteca was an early landrace during the flowering period, it was delayed in the next stadium (Figure 2b). No changes were recorded between the two treatments during flowering and pod development, except for Tortola. The apparent delay in Tortola during RW was related to the initial abort in the first flower. Changes among treatments were observed from physiological maturity days, showing in general an early maturity for RW for the landraces. Coscorron maintained the delaying behavior (Figure 2c). For the seed-filling days, a shorter period was reached by the landraces under RW, although no significant differences were detected between them, with the exception of Negro argel, which showed the shortest period (Figure 2d).

To identify relationships among the landraces that grouped phenological traits, we performed a K-means clustering analysis from individual replicates. The traits used were flowering days, pod days, physiological maturity days, and seed filling. The first cluster was composed of Coscorron, while Cluster 2 contained Blanco, Manteca, and a section of Negro argel (only the OW treatment). Finally, the third cluster was composed of Pinto, Enriqueta, Coyunda, Bayote, and a part of Tortola (OW treatment) (Figure 1 sup). Additionally, the clusters were visualized in a scatterplot matrix for the three clusters (Appendix A). Among them, the most clearly differentiated clusters were found for flowering and pod days, where Pinto and Coyunda showed early flowering and pod days, compared with Coscorron, which was very delayed. Blanco and Manteca were found in the middle cluster with intermediate values. Physiological maturity and seed filling did not show clear differentiation (Appendix A).

### 2.3. Growth Traits and Chlorophyll Concentration

Foliar area (FA) is an important trait that is reduced by the effect of drought. Here, Manteca, Tortola, and Pinto had lower values under RW, while Enriqueta, Bayote, Negro argel, Coyunda, and Coscorron showed higher values. All of them were reduced in a proportion of 38–56% during RW treatment (Figure 3a). To determine the growth ability in landraces and the effect of drought treatment, the relative rate of leaf area expansion (RLAE) was measured in a 60–74 DAS period in addition to FA. This parameter was significantly higher in OW than in RW (Figure 3b). Blanco and Bayote had a higher RLAE under OW conditions; however, they were greatly decreased by the RW treatment with 63% and 80% reductions, respectively. On the other hand, the landraces Tortola, Coyunda, and Enriqueta had lower RLAE values and did not show significant differences between the two treatments (Figure 3b). These landraces were between the lowest affected by the drought treatment.

The chlorophyll concentration was followed between 54 and 90 days during the course of the assay. This was considered to be from a flowering state until physiological maturity. Here, the chlorophyll concentration was reduced during the progress of plant maturation in both treatments; however, the values in RW were lower than those in OW (Figure 4a). The Bayote and Tortola landraces contained higher chlorophyll concentrations in OW, but they also had higher reductions, reaching similar values to the other landraces during RW. Similarly, Coscorron, Pinto, and Blanco showed a reduction in chlorophyll concentration during RW. Only Enriqueta showed a significant increase during RW (Figure 4b).

### 2.4. Yield Traits and % Reduction Rate Relationships based on the Effect of Water Treatment and Drought Tolerance Indices

The grain yield was highly affected by the water stress. A reduction of nearly 50% of yield was observed on average between both treatments. The yield reached the lowest value in Negro argel and Pinto and the highest in Manteca and Bayote under OW and comparatively between the landraces. Nonsignificant changes were found between Negro argel, Pinto, Tortola, Coyunda, and Enriqueta under RW, while Bayote, Coscorron, and Manteca were the landraces with the highest yield, although only significantly so with respect to Tortola and Negro argel (Figure 5a). Additionally, high variability was found for one hundred seed weight (P100). The values for the landraces changed between 21 and 59 gr. However, no significant differences were observed between the water treatments. The landraces Bayote and Enriqueta showed the highest value, while Negro argel showed the lowest P100 (Figure 5a). Additionally, we evaluated the parameter pod harvest index (PHI), which has been described to be important for predicting tolerance to drought stress. Lower values were found for Blanco and Enriqueta in OW than for Manteca, Pinto, and Negro argel. (Table 1). Additionally, Pinto, Manteca, and Negro argel showed a lower reduction in PHI under RW. In contrast, Coyunda was the most highly reduced landrace in PHI value under RW compared with OW.

The number of seeds by pod was between 2.7 and 5.4 but was reduced in general for all landraces during RW compared with OW, except for Coscorron and Bayote (Table 1). The reduction was from 19 to 36% under RW. A higher value of seeds by pod was found in Negro argel, which is coincident with the lower value of P100.

On the other hand, the pod number was significantly reduced by the RW treatment, reaching 40% of the OW treatment. The pod number was higher in Blanco and Manteca under OW and lowest in Bayote, Pinto, and Enriqueta. Bayote and Enriqueta had lower pod numbers under RW (Table 1). There was a positive relationship between pod number and seed number between the landraces and treatments (R^2^ = 0.80, *p* < 0.001) (Figure 6a).

Because traits and their comparison did not adequately describe the relative effect of water treatment over them, we calculated “% trait reduction” as the proportion of reduction for each trait as an effect of water stress. Several correlations were obtained and described (Figure 6). The landraces Pinto and Coscorron were not considered for this analysis because the first was a determinate landrace and the second showed delayed phenological traits and did not appear to share the same mechanism for drought tolerance. Here, we found that the % pod reduction was negatively correlated with the % yield reduction (R^2^ = 0.94, *p* < 0.001) (Figure 6b) and positively correlated with the % RLAE reduction (R^2^ = 0.78, *p* < 0.001) (Figure 6c). In this sense, Coyunda had a severe yield reduction in RW compared with OW, contrary to Bayote and Blanco. However, Coyunda showed the lowest reduction in pod number, while Bayote and Blanco showed the highest reduction. On the other hand, the %N° pod reduction was not correlated with the % seed-by-pod reduction, although both reduced the values during RW (Table 1). However, the % N° seed reduction was positively correlated with % seed-by-pod reduction (R^2^ = 0.70, *p* < 0.01). Additionally, a high negative correlation was found between the % yield reduction, % RLAE reduction (R^2^ = 0.84, *p* < 0.0 1), and % seed-by-pod reduction (R^2^ = 0.73, *p* < 0.01) (Figure 6d,e).

Now, with the aim of identifying the most contrasting landraces for drought tolerance in the Chilean landraces, some related indices were calculated. Here, drought tolerance indices (DTIs), geometric mean productivity (GMP), and mean productivity (MP) were considered. Both DTI and GMP coincided with Manteca and Bayote being significantly different as the most tolerant and Negro argel as the most sensitive. Additionally, Pinto, Tortola, Enriqueta, and Coyunda showed lower DTI, GMP, and MP indices, with a range between 1.1 and 2.1 for DTI, 358 and 491 for GMP, and 396 and 545 for MP (Appendix A). The most tolerant landraces, Manteca, Bayote, Coscorron, and Blanco, reached DTI values of 2.45–3.38, GMP = 534–626 and MP = 554–677.

## 3. Discussion

In Mediterranean environments, common bean is mainly grown under irrigated conditions, and water stress can affect plant growth and grain yield across the entire growing season [17]. However, the water shortage effect is higher when it occurs in the reproductive stage [33,34]. This research simulated Chilean Mediterranean growing conditions, where lower water availability and higher temperature are expected in the flowering stage. Here, water stress conditions started at 54 DAS and were evidenced from 60 DAS impacting Ψπ, which coincided with higher environmental temperatures (Figure 1). Drought stress treatment (RW) also greatly impacted the leaf temperature compared with OW, which has been described previously and associated with the inability to cool the plant through transpiration because the stomata are closed [35,36]. Higher temperature affects pod and seed set pre- and post-anthesis, reducing pod and seed set and inducing pod abscission [34]. Drought stress under Mediterranean conditions worsens the environmental temperature, negatively influencing post-anthesis pod set and finally impacting crop yield [37,38].

Several authors have described phenotypic traits related to drought-tolerant landraces and their relationship with grain yield. All of them showed a drought yield improving focus [39,40]. From an ecological point of view, plants have selected assorted mechanisms to survive drought stress, and they are more or less successful. For wild species, this indicates adaptation to environmental conditions for specific habitats [41,42]. Seed collections conserved in germplasm banks are an opportunity to explore the mechanistic variability that constitutes different solutions for confronting adverse environmental conditions. For breeders and farmers, this success is related to crop yield, where drought stress is one of the main Mediterranean environmental conditions.

The Chilean landraces have not been classified previously by their drought tolerance because breeding has focused on yield potential under irrigated conditions. However, ancient *Phaseolus vulgaris*, as a result of interchange between prehispanic cultures, has been maintained and has perdured as the main Chilean race, while others have hybridized with Mesoamerican and Andean races, giving a diverse collection of Chilean landraces [21,43]. Among these accessions, the commercial types of Tortola, Bayo, Blanco, Negro argel, Manteca, Pinto, Coyunda, and Coscorron were evaluated in this research, and they represent four races from Andean and Mesoamerican pools (Appendix A). This research showed that the accessions Manteca, Coscorron, and Bayote had higher yields under optimal conditions. Although they were highly affected by the watering treatment, they were also able to reach a higher yield than the other accessions (Figure 5b). However, the primary aim of this research was to determine the variability found in Chilean common bean germplasm in response to drought and the most relevant traits involved in adaptative success for drought tolerance.

In this research, Pinto and Coscorron landraces showed significant opposite phenological times (pod, grain pod, and physiological maturity). Additionally, Coscorron showed delayed phenological times compared with the other landraces. However, the period of seed-filling days was maintained, suggesting that changes can be more associated with different photoperiod sensitivities [44,45]. Additionally, all landraces were indeterminates except Pinto. This evidence indicates that not all changes among the landraces are comparable. For the same reason, some analyses did not include the Coscorron and Pinto landraces.

Water treatments did not significantly affect the phenological stages of flowering or pod days (Figure 2). This was consistent with results obtained by Polania et al. [24] in a Middle American gene pool composed of elite lines with greater drought tolerance. Here, drought treatment started at flowering, but until physiological maturity, no change was evidenced between treatments. We suggest that the shorter seed-filling period during drought treatment was mainly due to early physiological maturity as an effect of water deprivation (Figure 2c,d). This has been previously reported in other studies [22,24,27]. This was probably the effect of accelerating the last phase of dehydration, when the seed lost water but not dry weight, and the embryo became metabolically quiescent [46]. Additionally, the PHI was higher in Manteca, Pinto, and Negro argel than in the other landraces (Table 1); however, these landraces did not have a lower reduction in yield. This can suggest that the PHI is not the only requirement for describing tolerance to drought in common bean, and other traits need to be considered in Mediterranean environments.

On the other hand, from flowering, the accumulation of chlorophyll was reduced in both treatments (Figure 4). Other species regulate pod development and reallocation of reserves during maturation, including leaf senescence [47]. However, the change in chlorophyll concentration between both treatments for some landraces could be associated with a strategy to survive drought stress. Similar results have been reported in some Mediterranean species and are related to plant photoprotection [48,49,50].

Although drought stress did not have a strong effect on the phenology of the reproductive stage, it showed a significant effect on grain yield, which is consistent with other studies [11,51]. The reduction in grain yield in the Chilean landraces was the cause of pod number and seed-by-pod reduction (Figure 5, Table 1), while the seed weight was not affected significantly by drought stress (Figure 5). In this sense, our results suggest that drought induces a sink reduction, causing a lower number of seeds but with a proper weight for reproductive success. Abiotic stresses such as high temperature and drought in early reproductive stages often result in failure of fertilization or abortion of seeds [52,53,54,55] and consequently a reduced yield. Additionally, stress causes reduced photosynthesis that negatively impacts the source strength for the fixation of carbohydrates, which is an additive to the transport limitation of photosynthates to sink organs [29,31].

The Chilean landraces have not been previously described according to their drought tolerance. Certain indices permit the categorization of drought tolerance in plants (Caballero et al.). Here, Bayote and Manteca were the most water tolerant landraces, while Negro argel was the most water sensitive (Appendix A). The grain yield under OW and RW was mainly considered, where the most tolerant landraces maintained a one-way higher yield under RW than under OW but also had a higher yield under OW (Figure 5b). Here, the landrace Pinto exhibited a different behavior because it was an earlier but also a determinate landrace, while Coscorron was the most delayed with respect to the other landraces.

This research also showed different relationships that accounted for the effect of drought on grain yield and yield components. We found a positive correlation among seed and pod number for the Chilean landraces (Figure 6a). On the other hand, the yield reduction due to the effect of water treatments was negatively related to pod reduction (Figure 6b). Additionally, the positive relationship between the % N° seed reduction and % seed-by-pod reduction was based on the effect of drought on the grain developing in the pod. In other words, these results suggest that drought differentially affects the pod number and embryo development in the pod for landraces with different grain yield abilities. In this sense, landraces with a lower effect on the yield mediated by drought have lower pod amounts than those with optimal watering. Similarly, the number of pods does not indicate a higher yield under water stress in Mediterranean environments for the Chilean landraces. During the process of seed development, including embryo growth, seed filling, and seed metabolic activity, the presence of water is a key factor [56]. Genetic evidence indicates that reproductive structures have hydraulic control of expansive growth that affects abortion under mild water stress during flowering, while carbon starvation is the main factor inducing abortion during the most severe post-flowering drought [57]. This research suggests that seed abortion in the pod is the relevant factor related to yield reduction during drought, which is determinate in the early phase of the flowering stage. Therefore, although a reduction in seed number under drought is associated with less seeds by pod, landraces that maintain a higher grain yield under drought stress, compared with optimal irrigation, have lower seed abortion and maintain more seeds by pod.

The RLAE and FA were variable traits among Chilean landraces (Figure 3). We found a higher positive correlation between % RLAE reduction and % pod reduction and a negative correlation of % yield reduction with % RLAE reduction at the reproductive stage (Figure 6c,d). Among the landraces, Bayote and Blanco showed higher RLAE and FA under OW. These traits were severely affected for both landraces under RW. In contrast, Coyunda showed higher FA during OW, although it had a modest RLAE and the lowest effect under drought (Figure 3). Here, Coyunda was the landrace with the highest yield reduction under drought, in contrast to Bayote and Blanco. Here, the yield was not necessarily related to the source size because Bayote and Coyunda were similar in leaf area (Figure 3). However, the RLAE appears to be a key factor in grain yield, where Bayote was highly delayed during RW compared with OW, in contrast to Coyunda. Evidence from other researchers has demonstrated that a conservative leaflet growth strategy mediated by water availability during early plant development has a lower effect on yield [29,58]. This suggests that in contrast to the effect observed under the vegetative state, during the reproductive phase, sink strength is increased for indeterminate landraces (new leaves, flowers, and pods). Therefore, resources must be divided between leaf and pod development. Accordingly, the reduction in the relative rate of leaf area expansion (% RLAE reduction) is most severe in landraces with lower yield inhibition, probably promoting higher resource translocation for pod development than leaves, consecutively improving the yield under drought stress. Other researchers have described that under optimal conditions, leaf expansion is affected by the growth and conditions of other organs in the plant. When the leaf reaches maturity, growth stops, but it can start again if the other shoots are decapitated [59]. This can reinforce the statement that resources are distributed for the organ in development depending on several factors, including resource availability, competition ability, and genetic conditions.

On the other hand, the cell wall is responsible for restricting cell and leaf expansion. It is mediated by enzymes such as peroxidases, xyloglucan endotransglucosylases/hydrolases (XTHs), and expansins. Peroxidases promote the crosslinking of cell wall components and favoring stiffening, while XTH and expansin produce loosening of the cell wall. This is related to external factors such as light or water availability. Additionally, growth regulators such as exogenous gibberellin, brassinosteroids, or cytokinin stimulate leaf expansion, while abscisic acid is an inhibitor. This suggests that the way to accelerate or reduce the leaf rate of growth can be mediated by these mechanisms [60].

As a consequence of drought, the water availability for organ growth is limited, causing loss in pod set and reduced leaf development [61,62]. It impacts the water potential in pods and leaves; however, as a product of accumulative reserves in pods, it can be more negative than that found in leaves, generating a water flux to pods [31,61,63]. Similarly, these studies found a lower effect on yield for landraces that hardly delayed leaf growth over time.

We suggest that under a scenario of water deficit where stomatal conductance is reduced and consequently photosynthesis is reduced, this delay in RLAE implies a lower expense for the plant. This delay in RLAE implies a lower estimate of resources for leaf development. Consequently, the resources are redirected with higher strength to pod development, promoting a higher yield under drought. In this sense, lower leaf resources have to be translocated to pods prior to leaf senescence. This leaf stunting can also contribute to reducing leaf water loss by their lower leaf area compared with genotypes that quickly reach the final leaf area. Similar results have been found in indeterminate Durango landraces [31,64].

In conclusion, under a Mediterranean environment, water stress severely affects grain yield in common bean crops; however, different behaviors can be observed in Chilean landraces. These are related to phenological traits, differential pod and seed production, and relationships with leaf expansion during the reproductive state. These findings will be important for identifying genetic traits selected by domestication to Mediterranean environments and the improvement of crops to drought under these climatic conditions.

## 4. Materials and Methods

### 4.1. Plant Material and Experiment

Cultivars of common bean (*Phaseolus vulgaris* L.) were provided by the Germplasm Bank Network of the Instituto de Investigaciones Agropecuarias (INIA) under a standard material transfer agreement and an Institutional INIA Policy for ABS of plant genetic material following international agreement signed by Chile. The landraces used in this research are representative of different races according to the classification made by Paredes et al. [21] (Appendix A). They are present in the Chilean germplasm collection generated by Bascur and Tay [18]. The market classes of landraces considered here were Coscorron, Tortola, Manteca, Plain white (Blanco and Coyunda), Pinto, Small black (Negro argel), Bayo (Bayote), and Light Red Kidney (Enriqueta).

Experiment was conducted at the shelter facilities of INIA-Quilamapu Research Center, located in Chillán City (36°34′ S; 72°06′ O) in Chile. Shelters were 15 × 7 m and 3 m high rooms isolated with anti-aphids screen. The soil is an Andisol, commonly known as ‘trumao’ and is taxonomically described as medial over sandy skeletal, amorphic, thermic Humic Haploxerands [65]. The experimental period was extended from November 2020 to March 2021.

Seeds were established in plots that consisted of two rows 1 m in length using 45 cm between-row and 8.3 cm within-row spacing (12 plants m^−1^). Prior to sowing, the soil was fertilized with 80 kg ha^−1^ of N, 60 kg ha^−1^ of P_2_O_5_, 88 kg ha^−1^ of K_2_O, 44 kg ha^−1^ of S, and 36 kg ha^−1^ of MgO. Seeds were disinfected with Fludioxonile (Celest, Syngenta, South Swing Road Greensboro, NC 27409, USA) and Teflutrine (Force, Syngenta, South Swing Road Greensboro, NC 27409, USA). Optimal watering was applied until flowering. Management included weeding by hand.

Two experiments were conducted in different water environments, one under optimal watering (OW) conditions and the other under restricted watering (RW) conditions. Both experiments were established across from one another and separated by a 1-m wide strip. In the water isolation strip, a 1-m depth plastic film was installed for avoiding water flow from the OW experiment. A pressurized irrigation system with drip emitters (1 Lh^−1^, Netafim, Israel) was implemented in both experiments (OW and RW); one drip emitter per plant was considered.

In each experiment, the nine common bean landraces were arranged in a randomized complete block design with three replicates.

### 4.2. Soil Water Content

The soil water content was managed by the pressurized irrigation system. Volumetric soil water content (VSWC) was monitored with capacitance sensors (EC-5, Decagon, METER Group, Inc. Pullman, WA, USA, 2365 NE Hopkins Ct. Pullman, WA 99163, USA) at a 30 cm depth connected to a datalogger (EM 50, ECH_2_O, METER Group, Inc. USA, 2365 NE Hopkins Ct. Pullman, WA 99163, USA) to record data each hour. The water supply was set up in 64 L per plot once a week from sowing until 54 days after sowing (DAS). Afterward, the OW treatment was maintained at a VSWC ≥ 0.16 m^3^ m^−3^, which was equivalent to field capacity. The VSWC in the RW experiment was maintained at 0.8 m^3^ m^−3^ on average, which was equivalent to 20–30% of field capacity. At 76 DAS, the RW treatment was supplied with 64 L of water per plot with the aim of reaching the seed pod state.

### 4.3. Data Collection

The flowering date was measured in each replicate through a count of flowers by plants, considering the flowering time when a minimum of 50% of plants in the replicate were flowering. A similar method was utilized for days to pod and days to physiological maturity, where 50% of plants were 90% pod or yellowish pod, respectively. The leaf temperature was measured with an infrared thermometer (Fluke—574 Fluke, Corporation Inc. Everett, WA, USA, 6920 Seaway Blvd, Everett, WA 98203, USA) after the midday for each replicate once a week for a period of 5 weeks. The measurement was recorded in the middle section of the plant 50 cm above the foliage at a 90° angle to the ground. Cloudy days were not considered for measurement.

To quantify the foliar area (FA), two extended leaves by replicate were marked at the 5th node measured from the apex of the main stem; the central leaflet of the leaf was photographed 3 times during a 3-week period between 60 and 74 DAS. The images were calibrated using millimeter paper and analyzed using ImageJ software to obtain the leaf area expressed in cm^2^. To calculate the relative rate of leaf area expansion (RLAE), the natural logarithm of the leaf area of each replicate was used and graphed as a function of the days elapsed since the beginning of the experiment with each sampling period, obtaining the slope of the regression expressed in cm^2^ cm^−2^ day^−1^. To obtain chlorophyll concentration, two plants from each replicate were randomly selected, the central leaflet of a completely expanded leaf from the middle zone of the plant was selected, and its value was determined with a leaf green intensity meter (MC-100, APOGEE instruments, Inc.Logan, UT, USA, 721 west 1800 north, Logan, Utah 84321, USA). Because there is a direct correlation between the intensity of the green and chlorophyll content, the values were expressed in chlorophyll units (arbitrary units). A total of nine measurements were recorded between 54 and 90 DAS, and the final value of chlorophyll was considered the mean of the records of the evaluated period. The osmotic potential (Ψπ) was measured from 0.3 g of frozen leaf tissue, which was ground in a microtube. Aliquots of 10 µL were obtained for measurement in a VAPRO 5600 vapor pressure osmometer (Wescor, Logan, UT, USA). The results obtained as mmol kg^−1^ were transformed to MPa using the Van’t Hoff equation: MPa = (0.173 − (0.0269)(× mmol kg^−1^)) × 0.1.

For yield calculations, the seed harvest was conducted manually and staggered, considering the completely dry pod three times, at 1|115, 123, and 131 DAS. The total yield was considered the sum of the three harvested periods, as well as their components. Everything was adjusted to an area of 1 m^2,^ and the humidity of the seed was adjusted to 14%. The grain yield was obtained (g m^−2^) and the number of pods m^−2^, number of seeds m^−2^, number of seeds by pod, and 100 seeds were randomly selected per replicate to obtain the weight of 100 seeds (g) (P100). The pod harvest index (PHI; dry weight of the seed at harvest/dry weight of the pod at harvest × 100) was calculated as described by Beebe et al. [9]. The differences between water conditions for each landrace were calculated as ∆ = 100 − ((average of the landrace in RW/average of the landrace in OW) × 100), and the values were expressed in %.

Drought tolerance indices were calculated using the following relationships (Cabello et al. [66]):Drought tolerance index DTI = (Y_s_ × Y_p_)/(Y*_p_)^2^
Geometric mean productivity GMP = (Y_p_ × Y_s_)^0.5^
Mean productivity MP = (Y_p_ + Y_s_)/2
where Ys is the yield of a given genotype under restricted watering, Yp is the same but under optimal watering, and Y*_p_ is the mean yield of all genotypes under nonstress conditions.

### 4.4. Statistical Analysis

The data obtained were subjected to a combined analysis of variance (ANOVA). A mixed model was implemented in the JMP 9.0.1 software from SAS. The model considered the fixed effects of water conditions, landraces, and the water conditions landraces interaction. Replicates were considered as random effect. The mean comparison test was performed with the Fisher’s least significant difference (LSD) test, with a level 5% significance. The phenotypic relationship among the variables studied were evaluated through correlation analyses using the program JMP 9.0.1 from SAS.

## Figures and Tables

**Figure 1 plants-11-00749-f001:**
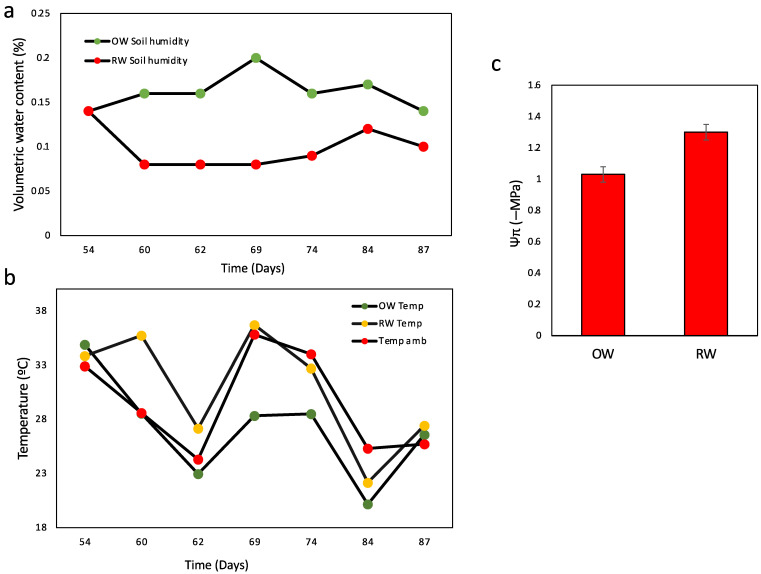
Water status and temperature conditions for the assay. (**a**) Soil water content evaluated over 33 days for two treatments, OW (green dot) and RW (red dot), at 54 DAS. (**b**) Environmental temperature (red dot) and leaf temperature for the OW treatment (green dot) and RW (yellow dot). (**c**) Osmotic potential in leaf under OW and RW at 60 DAS. Bars represent the mean of three biological replicates for each of nine landraces evaluated, while the error bars indicate the least significant differences (LSD) at *p* < 0.05.

**Figure 2 plants-11-00749-f002:**
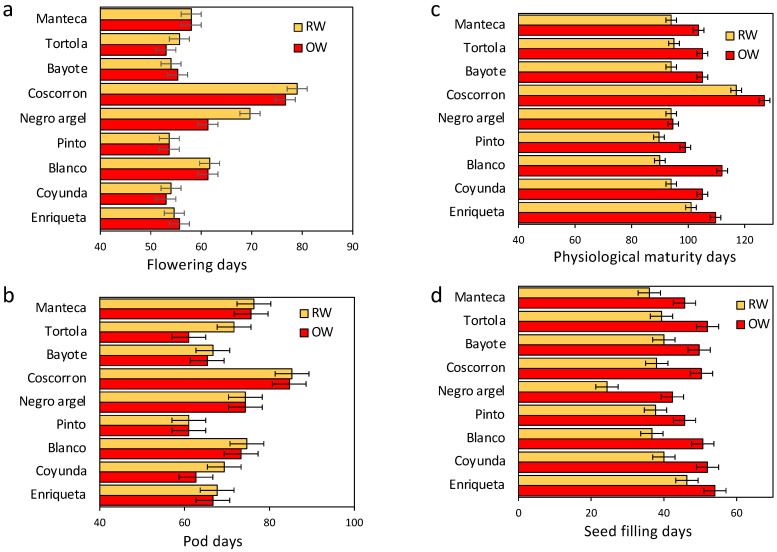
Effect of water stress on phenological traits for nine common bean Chilean landraces. Flowering days (**a**), pod days (**b**), physiological maturity days (**c**), and seed-filling days (**d**) under OW (red bar) and RW (orange bar). Bars represent the mean of three biological replicates, while the error bars indicate the least significant differences (LSD) value of the interaction water conditions by varieties at *p* < 0.05.

**Figure 3 plants-11-00749-f003:**
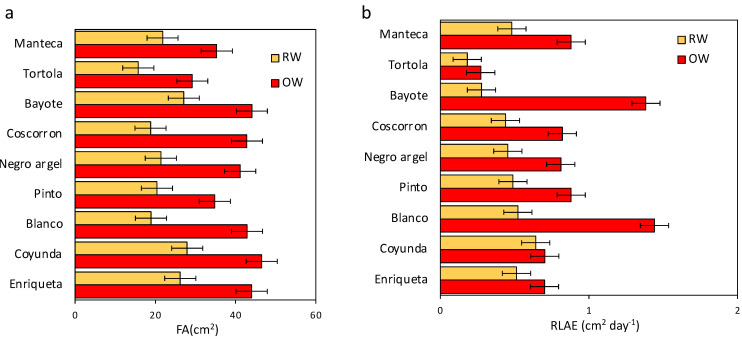
Effect of water stress on the rate of leaf growth for nine common bean Chilean landraces. (**a**) The relative rate of leaf area expansion (RLAE) was evaluated under OW (red bar) and RW (orange bar) treatments. (**b**) Leaf area expansion reduction corresponds to a reduction rate of RLAE in RW with respect to OW. Bars in (**a**) represent the mean of three biological replicates, while the error bars indicate the least significant differences (LSD) value of the interaction water conditions by varieties at *p* < 0.05.

**Figure 4 plants-11-00749-f004:**
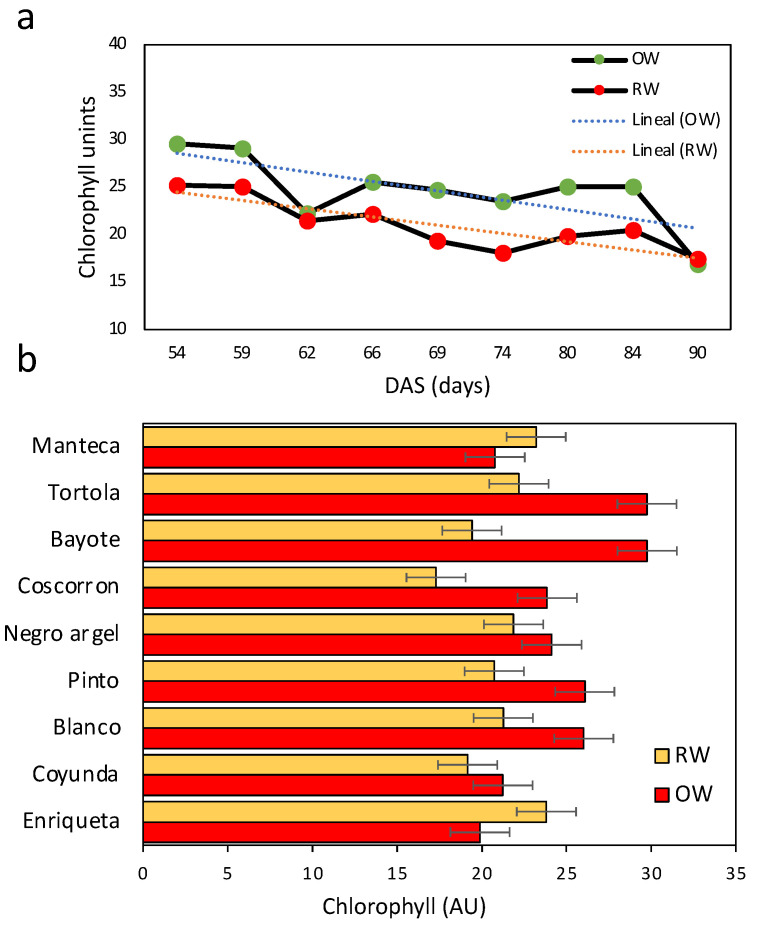
Effect of water stress on the concentration of leaf chlorophyll in common bean Chilean landraces. (**a**) Mean chlorophyll accumulation for nine common bean landraces under OW (green dot) and RW (red dot). (**b**) Chlorophyl accumulation in nine landraces under OW (red bars) and RW (orange bars). Bars represent the mean of 9 measurements during 54–90 DAS. Error bars represent least significant differences (LSD) value of the interaction water conditions by varieties at *p* < 0.05.

**Figure 5 plants-11-00749-f005:**
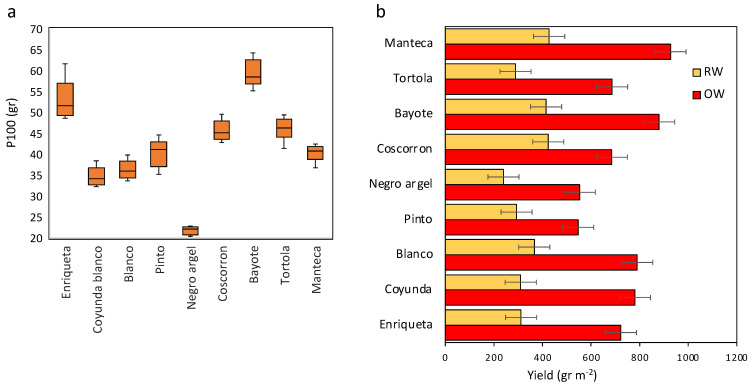
Effect of water stress on the yield and seed weight of Chilean common bean landraces. (**a**) One hundred seed weight and (**b**) Yield for the nine common bean landraces. Error bars represent least significant differences (LSD) value of the interaction water conditions by varieties at *p* < 0.05.

**Figure 6 plants-11-00749-f006:**
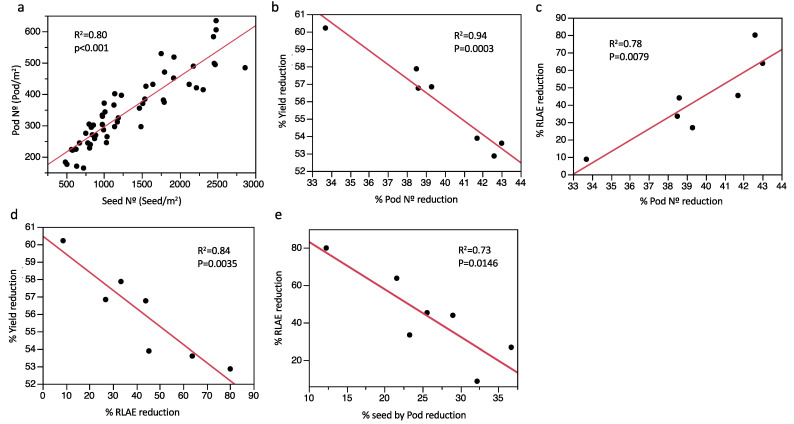
Relationship between (**a**) pod number (Pod N°) with seed number (Seed N°), (**b**) % yield reduction and (**c**) % RLAE reduction with % pod number reduction, (**d**) % yield reduction with % RLAE reduction, (**e**) % RLAE reduction with % seed-by-pod reduction. For (**a**), the dot of the graph represents replicates from each landrace. For the rest, the dot represents the calculated relationship for each landrace minus Coscorron and Pinto. A linear fit is shown as a red line. The regression coefficient (R^2^) and significance are also shown.

**Table 1 plants-11-00749-t001:** Effect of water stress on the traits Pod N°, Seed N°, Seed by pod, and Pod harvest index (PHI).

Landrace	Pod N° (Pod/m^2^)		Seed N° (Seed/m^2^)		Seed by Pod		PHI (%)	
	OW	RW	OW	RW	OW	RW	OW	RW
Enriqueta	340.4	206.7	1457.3	553.0	4.2	2.7	67.5	64.0
Coyunda	451.3	299.2	2136.6	945.4	4.8	3.2	74.2	65.6
Blanco	572.1	325.8	2227.7	987.7	3.9	3.0	69.5	66.3
Pinto	352.5	254.6	1281.8	767.7	3.7	3.0	79.4	77.4
Negro argel	465.8	286.3	2518.4	1108.7	5.4	3.8	78.8	76.5
Coscorron	437.9	290.8	1492.5	948.4	3.4	3.3	74.4	81.2
Bayote	353.3	202.9	1479.3	707.1	4.2	3.6	72.2	68.5
Tortola	390.4	240.0	1429.4	663.9	3.6	2.8	75.7	71.5
Manteca	520.8	303.8	2353.9	1037.9	4.6	3.4	79.5	75.9
*p* ≤ 0.05	*	ns	*	*	ns	*	**	***
LSD	114.2	-	600.4	294.0	-	1.0	4.6	3.4
Mean	431.6 a	267.8 b	1819.7 a	857.8 b	4.2 a	3.2 b	74.6 a	71.9 a

LSD values correspond to the least significant Fisher differences, * = *p* < 0.05; ** = *p* < 0.01; *** = *p* < 0.001; and ns = not significant. Different lowercase letters between columns indicate differences between water conditions.

## Data Availability

The data presented in this study are available in article and Appendix A.

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
