# Peer review of "Water Shortage Affects Vegetative and Reproductive Stages of Common Bean (Phaseolus vulgaris) Chilean Landraces, Differentially Impacting Grain Yield Components"

_plants, 2022, doi:10.3390/plants11060749_

Round 1

Reviewer 1 Report

I accept manuscript after corrections. The added excerpts improved the quality of the manuscript. I recommend for consideration.

Reviewer 2 Report

The authors have made all changes that were pointed out previously. Therefore, the manuscript can be accepted in its current form. 

This manuscript is a resubmission of an earlier submission. The following is a list of the peer review reports and author responses from that submission.

Round 1

Reviewer 1 Report

The title should be revised.

The authors should provide some more information about the methodology in the abstract section.

I will suggest the authors to start their introduction with the importance of legumes firstly for sustainable agriculture

Line 40-41: To support their statement, authors should include this study (doi.org/10.3390/genes11010036). 

As this study belongs to Chile, I will suggest the author's to provide some detailed information about the status, production, and limitation to common bean production in Chile. It will improve the readers understanding.

Aim and objectives are not clear, authors are suggested to clearly present the need of this study and how this study will improve our understanding at the end of the introduction section.

Line 99-102 belongs to  M&M

I will suggest the authors to perform correlation analysis

The discussion section can be shortened.

Author Response

Comments and Suggestions for Authors

1-The title should be revised.

Answer: Title was revised

2-The authors should provide some more information about the methodology in the abstract section.

Answer: thanks, we provide additional text.

3-I will suggest the authors to start their introduction with the importance of legumes firstly for sustainable agriculture.

Answer: We modified the start of introduction with the requested information.

4-Line 40-41: To support their statement, authors should include this study (doi.org/10.3390/genes11010036).

Answer: The study was added.

5-As this study belongs to Chile, I will suggest the author's to provide some detailed information about the status, production, and limitation to common bean production in Chile. It will improve the readers understanding.

Answer: We added a revision from Chilean common bean from Vagisnki et al.

6-Aim and objectives are not clear, authors are suggested to clearly present the need of this study and how this study will improve our understanding at the end of the introduction section.

Answer: Thanks, the aim was clarified

7-Line 99-102 belongs to  M&M

Answer: The paragraph was removed and added to M&M

8-I will suggest the authors to perform correlation analysis

Answer: We explored correlation analysis and we included the most important relationships among variables, which are showed in figure 6.

9-The discussion section can be shortened.

We think important maintain the extension of discussion.

Reviewer 2 Report

  1. In the part of “phenological aspect”, flowering days, pod days, physiological maturity days and seed filling days under OW and RW were compared. The citation relationship of results in the text should be listed one by one in the text (the current version listed fig.2a and fig.2d only).
  2. L129-131, It's hard to understand the sentence “Not changes were recurrently registered between the treatments in the landraces, excepting for Tortola, which moving forward during OW compared with RW.” If the results indicate there is no significant changes between OW and RW, which exception should be Nefro argel based on Fig.2a.
  3. The unit of the foliar area (FA) is suggested to replace “cm2” with “cm2/plant?” in Fig. 3a.
  4. The unit of the grain yield is suggested to replace “gr” with “gr/plant?” in Fig. 5b.
  5. It is not suitable for linear correlation analysis between “% seed by pod reduction” and “% seed no. reduction” of Fig.6f, due to the data is non-uniform distribution.

Author Response

Thank so much for your revision. Here you can find the answer to the question:

  1. In the part of “phenological aspect”, flowering days, pod days, physiological maturity days and seed filling days under OW and RW were compared. The citation relationship of results in the text should be listed one by one in the text (the current version listed fig.2a and fig.2d only).

Answer: It was corrected.

  1. L129-131, It's hard to understand the sentence “Not changes were recurrently registered between the treatments in the landraces, excepting for Tortola, which moving forward during OW compared with RW.” If the results indicate there is no significant changes between OW and RW, which exception should be Nefro argel based on Fig.2a.

Answer: The paragraph was clarified

  1. The unit of the foliar area (FA) is suggested to replace “cm2” with “cm2/plant?” in Fig. 3a.

Answer: The foliar area measurement was explained in M&M. It has to be expressed in cm2.

  1. The unit of the grain yield is suggested to replace “gr” with “gr/plant?” in Fig. 5b.

Answer: The foliar area measurement was explained in M&M. It has to be expressed in cm2.

  1. It is not suitable for linear correlation analysis between “% seed by pod reduction” and “% seed no. reduction” of Fig.6f, due to the data is non-uniform distribution.

Answer: We totally agree with the reviewer comment, however we used this figure referentially. It was important for us to show that there is a cline among common bean accessions for this relationship. In this sense, we would like to keep figure 6.

Reviewer 3 Report

Comment to manuscript 1469858 entitled "Water shortage affects vegetative and reproductive stages of common bean (Phaseolus vulgaris L.) Chilean landraces impacting differentially the grain yield components“.

This manuscript describes the phenotypic behavior of nine representative Chilean common bean landraces. I think that the subject is interesting and important, particularly from the practical point of view.

I should like to suggest a few improvements in the manuscript:

There is a lack of deep physiological discussion.

for example

Line 285. Water treatments did not affect significantly phenological stages of flowering days, pod days and days to physiological maturity. This sentence needs to be better explained physiologically (expand the explanation).

Line 362 We suggest that under the scenario of water deficit where stomatal conductance is reduced and consequently the photosynthesis, this delay in RLAE imply lower expense for the plant, that can contrarrest the oxidative damage, osmotic regulation, between others. This sentence needs to be better explained physiologically.

Explain the increase drought stress treatment by cell wall stiffening which can be regarded as a defence mechanism.

Description of the methods is imprecise.

Line 385 What does Triomag mean, what kind of fertilizer it is, describe please.

Line 420 its value was determined with a chlorophyll concentration meter (….) the values were expressed in SPAD. Explain in method:  SPAD is not the chlorophyll content, but the intensity of the green color in the leaves.

The figure need to be corrected: missing a, b, or c and d in figures 2, 3 4 and 5. Do not use the abbreviations in the figure caption.

Figures 3, 4 and 5. The figure a is a different size than figure b.

In the Figure 5. The variation names should be in the same order.

Have you noticed tissue dehydration ? What about specific weight ?

Needs improvement , because one time is p <0.05, another time is P ≤ 0.05.

Omit a full stop at end of sentence in paragraphs.

Also, Ms contains misprints, mistakes in English grammar and in the writing style. I recommend that the authors should use some help of a native English speaker or send the Ms to an English Editing Service that proofreads scientific writing.

The article should be published, however, basic shortcomings must be removed.

Author Response

Dear Reviewer, thank so much for your comments. Here you can find answer to your questions:

  • Line 285. Water treatments did not affect significantly phenological stages of flowering days, pod days and days to physiological maturity. This sentence needs to be better explained physiologically (expand the explanation).

Answer: More detailed were included and improve explanation.

  • Line 362 We suggest that under the scenario of water deficit where stomatal conductance is reduced and consequently the photosynthesis, this delay in RLAE imply lower expense for the plant, that can contrarrest the oxidative damage, osmotic regulation, between others. This sentence needs to be better explained physiologically.

Answer: Here more detailed were included and improve explanation.

  • Explain the increase drought stress treatment by cell wall stiffening which can be regarded as a defence mechanism.

Answer: Thank so much, we added a paragraph in discussion and additionally others factor involved in leaf expansion.

Description of the methods is imprecise.

  • Line 385 What does Triomag mean, what kind of fertilizer it is, describe please.

Answer: It was corrected

  • Line 420 its value was determined with a chlorophyll concentration meter (….) the values were expressed in SPAD. Explain in method:  SPAD is not the chlorophyll content, but the intensity of the green color in the leaves.

Answer: It was improved and clarified

  • The figure need to be corrected: missing a, b, or c and d in figures 2, 3 4 and 5. Do not use the abbreviations in the figure caption.

Answer: It was corrected

  • Figures 3, 4 and 5. The figure a is a different size than figure b.

Answer: Here was corrected the width of figure 4, width and heigh of figure 3 and position and order of figures 3,4 and 5. We think that answer to the reviewer request.

  • In the Figure 5. The variation names should be in the same order.

Answer: It was corrected

  • Have you noticed tissue dehydration ? What about specific weight ?

Answer: RWC was not measured. Osmotic potential was measured, however the data not contribute to improve the research and not was included here.

  • Needs improvement, because one time is p <0.05, another time is P ≤ 0.05.

Answer: It was corrected.

  • Omit a full stop at end of sentence in paragraphs.

Answer: The research was submitted to English edition.

  • Also, Ms contains misprints, mistakes in English grammar and in the writing style. I recommend that the authors should use some help of a native English speaker or send the Ms to an English Editing Service that proofreads scientific writing.

Answer: It was subject to English review by editing service.

Round 2

Reviewer 1 Report

Manuscript can be accepted in its current form

Reviewer 2 Report

The revised version had made some improvement but still failed to match the requiement for publication of the journal. I provided 5 comments for reviewing, but there are 3 comments(#3-5) without any responses.

Reviewer 3 Report

The manuscript has been corrected and I accept this version.